# Mechanical Properties and Serration Behavior of a NiCrFeCoMn High-Entropy Alloy at High Strain Rates

**DOI:** 10.3390/ma13173722

**Published:** 2020-08-23

**Authors:** Ruoyu Liu, Xianrui Yao, Bingfeng Wang

**Affiliations:** School of Materials Science and Engineering, Central South University, Changsha 410083, China; lry2019@csu.edu.cn (R.L.); yaoxianrui@csu.edu.cn (X.Y.)

**Keywords:** high-entropy alloy, mechanical properties, strain rate, serration behavior

## Abstract

Serration behavior is a kind of plastic instability phenomenon of materials, which widely exists in the high-entropy alloys and has influence on microstructure and mechanical properties. In this work, the microstructure and mechanical properties of a NiCrFeCoMn high-entropy alloy (HEA) were studied under high-speed impact. The microstructure of a NiCrFeCoMn HEA were investigated by optical microscope (OM), scanning electron microscope (SEM), electron backscatter diffraction (EBSD), and transmission electron microscope (TEM). The dislocation density increased with the true strain at high-strain-rate deformation, and the dislocations can be hindered and released continually by the twin layers, resulting in serration on the true stress—true strain curve. When values of the strain rates are 1250, 2000 and 4800 s^−1^, the yield strength of the deformed NiCrFeCoMn HEA are 510, 525 and 680 MPa, respectively. Moreover, the fluctuation of the serration became more serious with the increasing of the strain rate. Compared with the as-cast NiCrFeCoMn HEA, the true stress—true strain curve of the deformed NiCrFeCoMn HEAwas smoother.

## 1. Introduction

High-entropy alloy (HEA) came to light at the beginning of this century and became a research hotspot rapidly with its unique properties [1]. NiCrFeCoMn high-entropy alloy, also known as Cantor alloy [2], is favored for its single-phase (face-centered cubic, FCC) [3] and mechanical properties [4,5,6,7,8,9]. Lots of studies investigate the mechanical properties of the alloy under quasi-static conditions (10^−3^ s^−1^–10^−1^ s^−1^); however, little attention has been paid to the mechanical properties of the materials under dynamic loading. As is known to all, the dynamic mechanical properties of materials are worthy of attention in the ballistic impact and penetration applications [10]. In addition, transportations and spacecraft under high-speed movement also need to be considered. Therefore, the dynamic mechanical behavior of the HEAs needs to attract considerable attention [11], and exploring the dynamic mechanical properties of the HEA may make it have a broader application.

The serration behavior usually presents a saw-like appearance on the stress–strain curves [12], which occurs in many structural and functional materials [13] and is unstable as well as not beneficial to further production and use of the material. Researchers have put forward some theoretical models [14,15,16,17,18,19] to demonstrate it, and the effect of Portevin-Le Chatelier (PLC) [20,21,22], referring to the existence of continuous serrated yielding on the stress–strain curve within a certain temperature and strain rate range, is adopted most to interpret the serration behavior in HEAs. The results of recent studies [23,24] suggest that there are serration behaviors of the NiCrFeCoMn HEA in the tensile and compressive tests. In addition, Wang et al. [25] have discussed the serration behavior in a NiCrFeCoMn HEA prepared by powder metallurgy and found that serration behavior is related to the strain rate. Moreover, the as-cast NiCrFeCoMn HEA was compressed under dynamic impact [26], and it found that there were serrations on the true stress–strain curves. However, the dendrites and grave element segregation in the as-cast alloy can seriously affect the stability of the mechanical properties of the alloy. Therefore, improving the homogeneity of the alloys may lighten the saw-tooth behavior on the true stress–strain curve. 

In this study, the homogenization process and multi-directionl forging experiments were used to refine grains of a NiCrFeCoMn HEA. The aims of this paper are as follows: (1) to investigate mechanical properties and microstructure of a deformed NiCrFeCoMn HEA under dynamic loading; (2) to obtain microstructure for the serration behavior of a NiCrFeCoMn HEA; (3) to discuss the microstructure mechanism of the serration behavior of a deformed NiCrFeCoMn HEA under high strain rates.

## 2. Materials and Methods 

The as-cast equiatomic NiCrFeCoMn high-entropy alloy was prepared by the induction melting under vacuum with an Ar atmosphere using high-purity metals, such as Ni, Cr, Fe, Co, Mn. The chemical composition of the NiCrFeCoMn high-entropy alloy is given in Table 1. The homogenization process was conducted for 2 h at 1200 °C and the surface of samples was covered with glass powder to keep the air out. Then, the samples with a size of 20 mm × 16.33 mm × 13.33 mm (1.5:1.22:1) were placed in the furnace with a temperature of 800 °C for 30 min before multi-direction forging. Forging was carried out at a compression rate of 20 mm/min for three passes, which was loaded along the X, Y and Z axis in turn. After each passes, the samples were quickly put into water for quenching and cooling and then were re-cut with a ratio of 1.5:1.22:1. After one cycle, which contains three passes forging, the total deformation, the sum of deformation in each direction of the sample, was 0.97. Quasi-static compressive tests and dynamic compressive tests were carried out by an INSTRON 8802 machine and a split-Hopkinson pressure bar (SHPB), respectively. A series of samples of different sizes for the dynamic testing were listed in Table 2. Because the smaller the sample size, the higher the strain rate for the dynamic tests. The strain rate, true strain and true stress for dynamic tests can be obtained by the following formulas.
(1)$$\dot{\varepsilon }=-\frac{2C_{0}}{L_{s}}\varepsilon_{r}\left(t\right)$$
(2)$$\varepsilon =-\frac{2C_{0}}{L_{S}}\int_{0}^{t}\varepsilon_{r}\left(t\right)dt$$
(3)$$\sigma =\frac{A_{0}}{A_{S}}E_{0}\varepsilon_{t}\left(t\right)$$
where *C*_0_ and *A*_0_ are the elastic wave speed and cross-sectional area of a SHPB, respectively; *E*_0_ is the elastic modulus of the bar; *L_s_* is the height of the sample; *ε_r_* (t) and *ε_t_* (t) represent the measured strain of incident and transmitted stress pulse by experiments, respectively.

The optical micrographs were carried out by optical microscope (OM, POLYVAR-METHMV-2, Denmark). The etchant for the alloy was 5 g CuSO_4_·5H_2_O + 25 mL hydrochloric acid + 25 mL water. Electron backscattered diffraction (EBSD) was tested by scanning electron microscope (SEM, ZEISS EVO MA10, Germany) under 20 kV. The EBSD data were analyzed by Channel 5 software, and a layer of Euler angle was added, in which the size of Kuwahar filter was 3-pixel points × 3-pixel points with a 5° smoothing angle. Additionally, transmission electron microscope (TEM, Tecnai G^2^ T20 ST, USA) was used under 200 kV.

## 3. Results

Figure 1 is the initial microstructure of the NiCrFeCoMn HEA. Figure 1a is the optical micrograph of the as-cast NiCrFeCoMn HEA. There are a lot of dendrites in Figure 1a, indicating the element segregation of the as-cast NiCrFeCoMn HEA. After the homogenization process, the shape of grains changes a lot, where the dendrites disappear and grains are polygon with the grain size ranging from 160–700 μm, shown in Figure 1b. Figure 1c is the optical micrograph of the deformed NiCrFeCoMn HEA. Compared with Figure 1b, the average dimension of grains in Figure 1c decreases to 125–450 μm with twins appearing, which are caused by plastic deformation during multi-direction forging. The sliding deformation alone is not enough to resist plastic deformation in the alloy, which leads to the coordination of twin deformation and plastic deformation, resulting in deformation twins. Figure 1d displays the bright field electron image of initial alloy, showing that the grain boundary is polygonal, which suggests that dynamic recrystallization occurs during deformation [27]. In addition, the dislocations in the deformed NiCrFeCoMn HEA are distributed randomly in the matrix while no parallel micro bands exist.

Figure 2 shows the compressive true stress—true strain curves of the deformed NiCrFeCoMn HEA at different strain rates. At room temperature (298 K), the compressive yield stress of material at quasi-static (0.001 s^−1^) is about 320 MPa, and the yield strength of the deformed NiCrFeCoMn HEA increases with the strain rates. When the strain rates are 1250, 2000, 4800 s^−1^, the dynamic yield stress is 510 MPa, 525 MPa and 680 MPa, respectively. In addition, the curve is smooth without clear serration behavior under the quasi-static deformation condition. Nevertheless, the higher the strain rates, the more serious serration behavior gradually appeared, indicating that the deformation process is more unstable with the increasing of the strain rates.

Figure 3 displays the optical micrographs of the deformed NiCrFeCoMn HEAs under different strain rates. Figure 3a–c are the microstructures of the deformed samples under 1250, 2000 and 4800 s^−1^, respectively. After the impact, the samples are deformed but not broken. Compared with the as-cast alloy (Figure 1a), it is evident that the grains are deformed after the high-speed compression, and the deformation degree of grains increases with the strain rates. When the strain rate reaches 4800 s^−1^ (Figure 3c), the shape of grains changes into a long strip.

Figure 4 is the EBSD images of the deformed NiCrFeCoMn HEA compressed at different strain rates. Figure 4a illustrates the Euler image of a deformed NiCrFeCoMn HEA impacted at the strain rate of 1250 s^−1^. It can be found that a number of micro bands exist at the grain boundaries. Zooming in grain boundary locally, as shown in Figure 4b, micro bands crisscross. Within each small region, micro bands are parallel to each other. Figure 4c,d display the Euler image of a deformed NiCrFeCoMn HEA at the strain rate about 2000 s^−1^. The high-angle grain boundary (60°) is marked by the black line. Compared with Figure 4a, the grains are refined and the fine twin layers are generated.

Figure 5 is the bright field electron images of the deformed NiCrFeCoMn HEA impacted at the strain rate 2000 s^−1^. As shown in Figure 5a, a number of parallel micro bands, which are composed of dislocations, appeared in the grains. Figure 5b shows that there is an interaction between dislocations, and a lot of dislocations can be seen within the deformation bands. In addition, the diffraction pattern of the internal structure of the micro band indicates it is a single FCC structure without a second phase. And no solid solution atoms are observed in the external microstructure of micro bands. Figure 5c shows that tangles even appear in areas with serious deformation. Zooming in the picture partially, the dislocation cells can be observed in Figure 5d.

## 4. Discussion

### 4.1. Dynamic Mechanical Properties of NiCrFeCoMn HEA

Figure 6 demonstrates the compressive true stress—true strain curves of the as-cast and deformed NiCrFeCoMn HEAs at different strain rates.

When the strain rate is about 0.001 s^−1^, the yield strength of the deformed sample is 320 MPa, which is higher than that of the as-cast alloy, 200 MPa. Reason for that is the grain size refines after multi-direction forging. When the strain rate is about 1250 s^−1^, the yield strength of the as-cast and the deformed samples are about 400 and 510 MPa, respectively, which is owing to fine-grain strengthening. Due to the better microstructure uniformity of the deformed alloy, the resistance of movable dislocation in the deformed specimen is less than that of the as-cast alloy, leading to a smaller periodic fluctuation of external stress. Thus, the fluctuation of the serration behavior of the deformed HEA is smaller than that of the as-cast alloy. When the strain rate reaches approximately 4800 s^−1^, the dynamic yield strength of the as-cast and the deformed samples are about 710 and 680 MPa, respectively. The research [28] indicates that for the same material at a given temperature, there is a transformation from uniform flow to inhomogeneous localized flow on the deformation of the sample with the increase of strain rate. When the strain rate reaches approximately 4800 s^−1^, the sample may be under the fluid state. Therefore, the yield strength of the as-cast sample is higher than that of the deformed sample, and the serration behavior of both samples are serious. Reason for that is the severe deformation leading to the high dislocation density in the microstructure of NiCrFeCoMn HEA. Then, stronger dislocation interaction and more severe local stress concentration bring about higher obstruction force in the process of motion, resulting in the required external stress for it far exceeding that of the samples at the other strain rates.

The serration fluctuation on the true stress–strain curve of deformed NiCrFeCoMn HEA is more severe compared with the powder metallurgy NiCrFeCoMn HEA [25], which is due to the finer equiaxed grains of the latter.

Therefore, for the same condition of the NiCrFeCoMn HEA, the serration presented on the compressive true stress–strain curve was more serious with increasing of the strain rate. When the strain rates are the same or similar, the fluctuation of the serration behavior of the as-cast alloy is higher than that of the deformed HEA. All of these results can attribute to the grain refinement with uniform microstructure and resistance of movable dislocation after forging.

Figure 7 shows the relationship between the yield strength and the strain rate of the as-cast and deformed NiCrFeCoMn HEAs. The defination of strain rate sensitivity (*m*) is presented as Formula (4):(4)$$m=\frac{d\left(log\sigma \right)}{d\left(\log\dot{\varepsilon }\right)}$$

The NiCrFeCoMn HEAs under different microstructure conditions both have high strain rate sensitivity at high strain rates, and the yield strength increases vary with the strain rate. However, the slope of yield strength of the deformed samples is less than that of the as-cast samples, and the *m* value of the as-cast and deformed NiCrFeCoMn HEA are 0.45 and 0.25, respectively, which indicates that the strain rate sensitivity of the deformed NiCrFeCoMn high-entropy alloy is lower than that of the as-cast sample.

### 4.2. Microstructure Mechanism for Serration Behavior of the Deformed High-Entropy Alloy at High Strain Rates

By studying and analyzing the serration behavior and microstructure of the deformed NiCrFeCoMn HEA, various pieces of evidence show that the micro bands in the samples after impact are closely related to the serration behavior on the stress–strain curve. The phenomenon shown in Figure 4 coincides with the experimental results of Zhang et al. [9]. The HEA mainly coordinates deformation through the dislocation slip, and micro bands parallel to each other will appear in the samples after impact. Couzinie et al. [29] found that slip bands became entangled with each other as the strain increases, which was confirmed in Figure 5c.

Figure 8 is the schematic diagram of the microstructure changes related to the serration behavior of the deformed NiCrFeCoMn HEA under high-speed compressive loading. Figure 8a displays the microstructure of the deformed NiCrFeCoMn HEA before the impact, where dislocations distribute randomly and a large number of twin layers exist. The dislocation density in the microstructure of the NiCrFeCoMn HEA changes while the alloy impacts under external loading, and the interaction between dislocations also changes at the same time. Figure 8b represents the infancy of dynamic impact deformation. The grains deform and are compressed along the impact direction, resulting in an increase in dislocation density in the grains. The dislocations start to move in order to coordinate the large deformation. However, the movable dislocations are obstructed by twin layers and pile up later, causing local stress concentration. As shown in Figure 8c, the dislocations later break away from the obstruction of the twin layers and continue to move under the action of external stress, leading to the reduction of the required external stress.

After the deformation process, the low energy storage micro bands are composed of sub-crystals, including movable dislocations. In the whole impact deformation process, the movement of movable dislocations is periodically hindered by lots of twin layers, leading to periodic fluctuations of external stress, which present as serration behavior. The PLC theory indicates that serration behavior is related to movable dislocation and the binding/breaking process of solute atoms, while the diffusion rate of solute atoms is relevant to deformation temperature and strain rate. For the same material, PLC critical strain, the onset of serration behavior on the flow–stress curve during the plastic deformation is linked to deformation temperature and deformation rate [20]. Kubin et al. [30] studied the evolution law of movable dislocations during deformation based on the effect of PLC. According to the theoretical analysis [30], the serration behavior of the deformed NiCrFeCoMn HEA also has critical strain at different strain rates.

Fu et al. [31] induced the relationship between the critical strain and the experimental strain rates. When the strain rate is greater than 1 × 10^3^ s^−1^,
(5)$$\varepsilon_{c}=\gamma {\left[\exp\left(Q/kT\right)\right]}^{3}T\dot{\varepsilon }+\varepsilon_{y}$$
where *γ* is the integrated coefficient; *Q* is the migration activation energy of solute atoms in the matrix; *k* represents the Boltzmann constant; *T* is the absolute experimental temperature; $\dot{\varepsilon }$ is the strain rate; and *ε_y_* is the yield strain. For the same material, all parameters are the same, so Formula (5) can be simplified to:(6)$$\varepsilon_{c}=\varphi \dot{\varepsilon }+\varepsilon_{y}$$
where φ is the integrated coefficient, a constant. When the sample is under high-speed impact loading, a large number of twin layers replace the solid solution atoms and block the motion of movable dislocations. Therefore, the simplified formula is used to fit the critical strain.

The result of fitting is that φ = 7.4 × 10^−7^. Figure 9 shows the comparison between the results obtained by formula calculation and the experiments when the loading strain rate is over 1 × 10^3^ s^−1^ for the deformed NiCrFeCoMn HEA. It can be found that the experimental and fitting results of critical strain both increase along with the increase in strain rates. In addition, there is a certain deviation between the experimental and the predicted results, yet the difference value is not large and the overall trend is consistent. As the strain rate increases from 1 × 10^3^ s^−1^ to 4 × 10^3^ s^−1^, the deformation velocity increases, while the binding/breaking of movable dislocation still needs some time. Therefore, the speed of breaking away from the obstruction is less than that of deformation, so that the critical strain value increases. The critical strain of the serration behavior of the deformed NiCrFeCoMn HEA also increases in pace with the increase in strain rates.

## 5. Conclusions

As the strain rate increases, the yield strength of the deformed NiCrFeCoMn HEA increases. When the strain rates are 1250, 2000, and 4800 s^−1^, the dynamic yield stress is 510, 525 and 680 MPa, respectively. Additionally, the serration behavior appears on the true stress–strain curve and becomes more severe. After high-speed impact, a number of micro bands appeared and are arranged in an approximately parallel manner within each zone. In addition, there are cutting and crossing and the other interaction phenomena among the micro bands, and entanglement appears in serious deformation areas. The deformation area is a single FCC structure without a second phase. At the same time, the dislocation movement process is hindered by the twin layers and cause local stress concentration, leading to the plastic instability, which is shown as the serration behavior on the mechanical curve. When the strain rate is over 1 × 10^3^ s^−1^, the critical strain of the serration behavior for the deformed NiCrFeCoMn HEA gradually increases with the strain rates. Finally, due to the better microstructure uniformity of the deformed alloy, the fluctuation of the serration behavior of the deformed HEA is smaller than that of the as-cast alloy.

## Figures and Tables

**Figure 1 materials-13-03722-f001:**
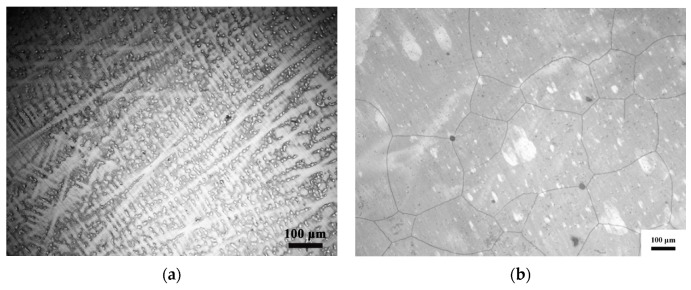
Initial microstructure of the NiCrFeCoMn HEA. (**a**–**c**) Optical micrographs for the as-cast sample and the alloy after homogenization heat treatment and the deformed alloy, respectively. (**d**) Bright field electron image for the deformed NiCrFeCoMn high-entropy alloy.

**Figure 2 materials-13-03722-f002:**
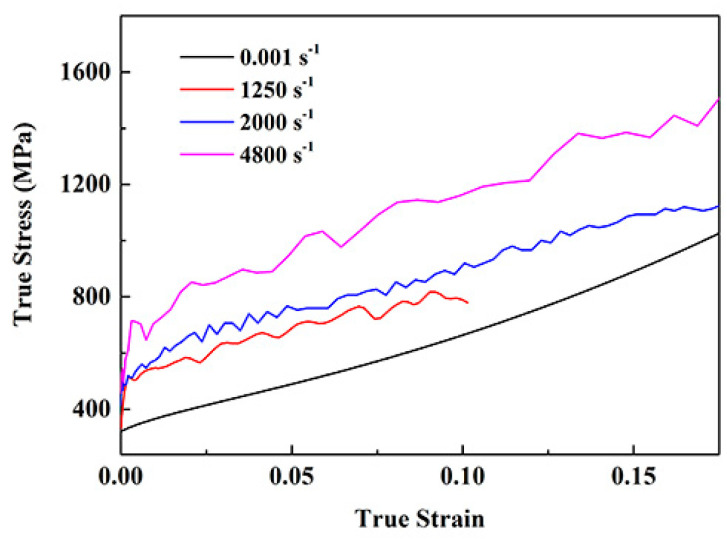
Compressive true stress vs. true strain curves of the deformed NiCrFeCoMn HEA at different strain rates.

**Figure 3 materials-13-03722-f003:**
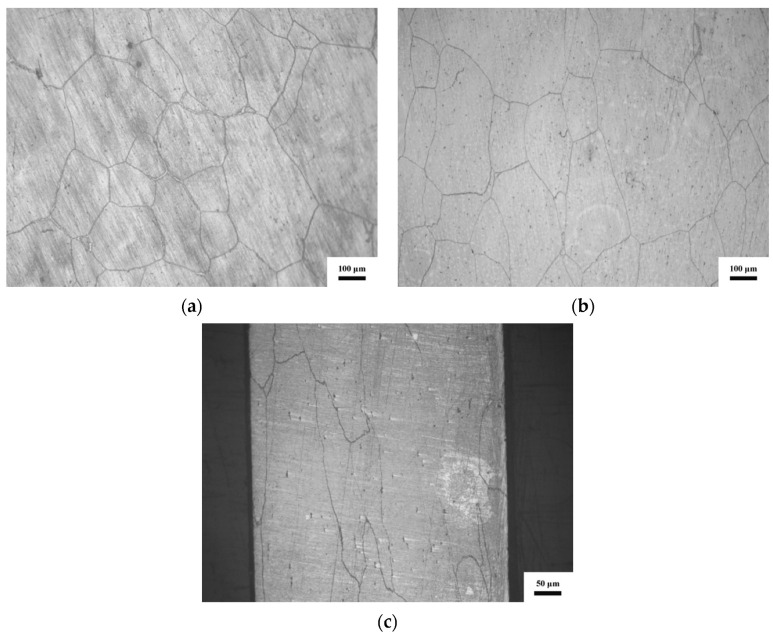
Optical micrographs of deformed samples under different strain rates: (**a**) 1250 s^−1^; (**b**) 2000 s^−1^; (**c**) 4800 s^−1^.

**Figure 4 materials-13-03722-f004:**
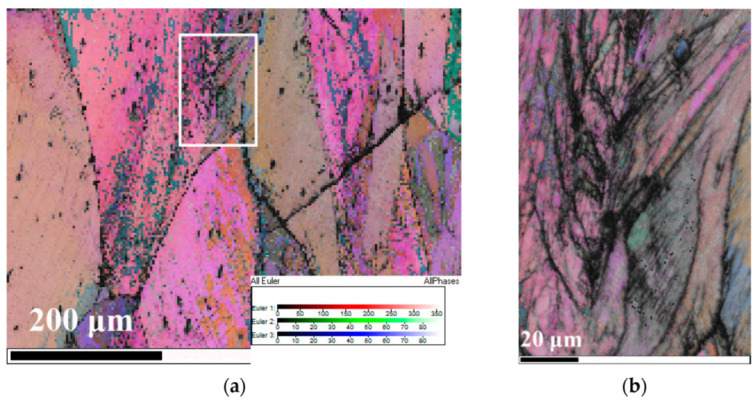
Electron backscattered diffraction images of the deformed NiCrFeCoMn HEA: (**a****)** Euler+Band Contrast (BC) image of the deformed sample at the strain rate 1250 s^−1^; (**b**) image of the micro bands in (**a**); (**c**) Euler+BC image of the deformed specimen at the strain rate 2000 s^−1^; (**d**) image of the twin band in (**c**).

**Figure 5 materials-13-03722-f005:**
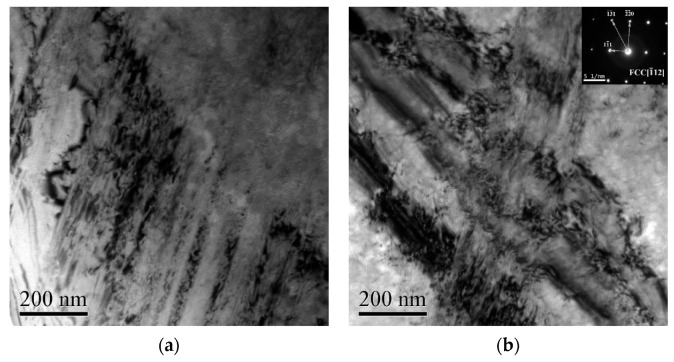
Bright field electron image of the deformed NiCrFeCoMn HEA impacted at the strain rate 2000 s^−1^. (**a**) Microbands; (**b**) microbands interacting with each other; (**c**) dislocation entanglement; (**d**) dislocation cell.

**Figure 6 materials-13-03722-f006:**
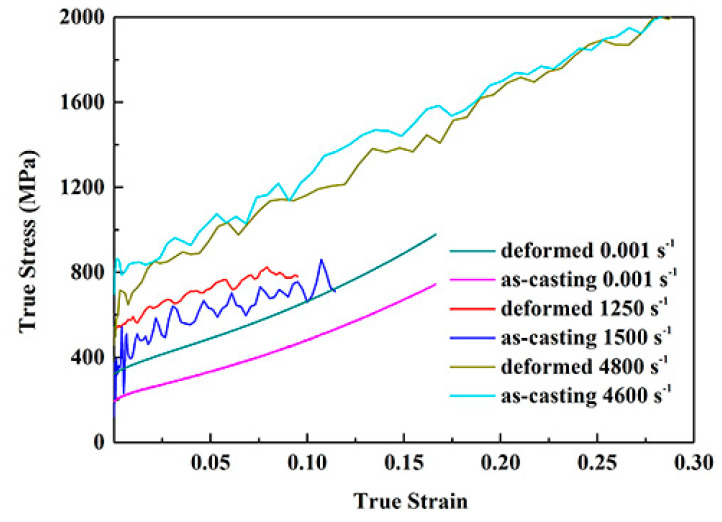
Compressive true stress–strain curves of NiCrFeCoMn HEA under different status at different strain rates.

**Figure 7 materials-13-03722-f007:**
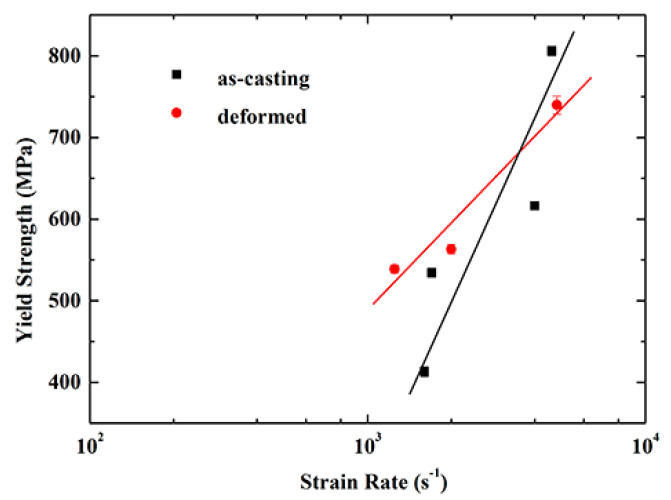
Comparison of yield strength vs. strain rate curves between the as-cast and deformed NiCrFeCoMn HEA.

**Figure 8 materials-13-03722-f008:**
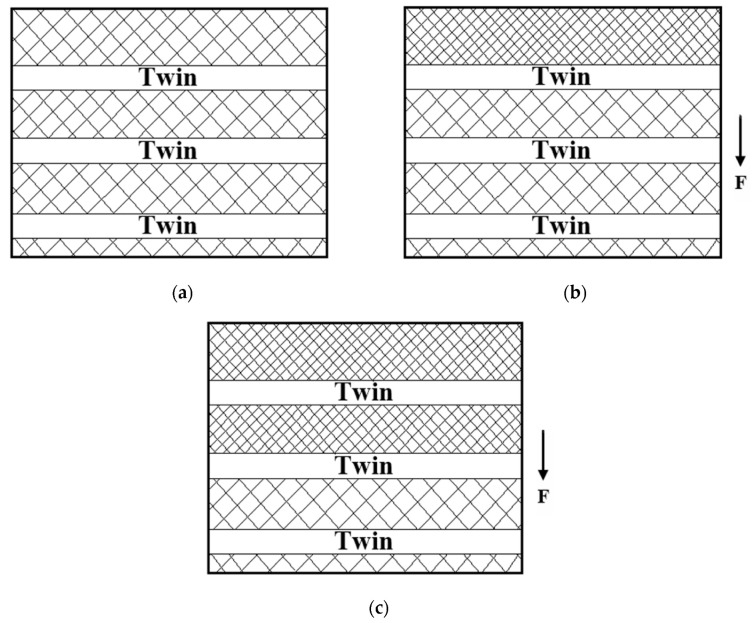
Schematic images for the microstructure change of the serration behavior in the deformed NiCrFeCoMn HEA. (**a**) Twin layers in the deformed sample; (**b**) the movement of dislocations is hindered by the twin layers; (**c**) dislocations overcome the twin layers.

**Figure 9 materials-13-03722-f009:**
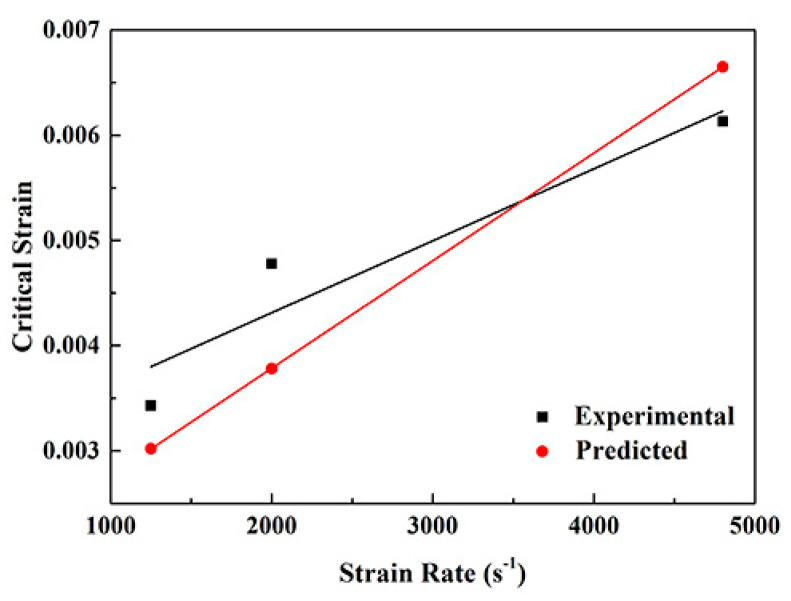
Comparison between predicted and experimental results of critical strain.

**Table 1 materials-13-03722-t001:** Chemical composition of the NiCrFeCoMn high-entropy alloy (HEA).

Elements	Ni	Cr	Fe	Co	Mn
at.%	21.24	17.76	21.79	20.72	18.49
wt.%	22.18	16.42	21.64	21.71	18.06

**Table 2 materials-13-03722-t002:** Sizes and the compressive strain rates of the samples.

Sample	Condition	Size (mm)	Compressive Strain Rate (s^−1^)
1	as cast	Φ4 × H5.6	0.001
2	as cast	Φ4 × H5.6	1500
3	as cast	Φ2 × H2.8	4600
4	deformed	Φ4 × H5.6	0.001
5	deformed	Φ6 × H8.4	1250
6	deformed	Φ4 × H5.6	2000
7	deformed	Φ2 × H2.8	4800

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
