# Peer review of "Mechanical Properties and Serration Behavior of a NiCrFeCoMn High-Entropy Alloy at High Strain Rates"

_materials, 2020, doi:10.3390/ma13173722_

Round 1
Reviewer 1 Report
In this paper, the mechanical properties of a NiCrFeCoMn high-entropy alloy were measured at various strain rates. Based on the analysis of microstructures, the authors tried to explain about a serration behavior on stress-strain relations. I recommend publication on ‘Materials’ after minor modifications based on the attached file.

Author Response
Thanks for the reviewer’s valuable suggestions. The response to the comments is attached.

Reviewer 2 Report
The paper shows the scientific quality to be accepted in the journal, together with the important research results in the field of High Entropy Alloys. Before the acceptance in the Journal, the following minor revision should be considered.
(1) The size of scale bar including font size in pictures is not uniform. Some scale bars are too small to read. Please unify the size of scale bar including font size.
(2) Short comments about the possibility of the formation of precipitates during deformation should be added in the revised paper. When there are not precipitates in the present work, the experimental results should be explained in the revised paper.
Author Response
Thanks for the reviewer’s helpful comments. The scale bar is not uniform for better comparison of pictures. We have modified the font size in the pictures. In addition, no precipitates was found in this work, which may be related to the composition and structures of this material. We will further study the possibility of the formation of precipitates during deformation in the future.
Reviewer 3 Report
The paper is interesting and well organized. The subject is worthy of investigation and the experimental setup is correct. The paper is publishable as it is.
Author Response
Thanks for the reviewer’s comments.